# Application of Iodine as a Catalyst in Aerobic Oxidations: A Sustainable Approach for Thiol Oxidations

**DOI:** 10.3390/molecules28196789

**Published:** 2023-09-24

**Authors:** Lijun Wang, Lingxia Chen, Zixuan Qin, Ke Ni, Xiao Li, Zhiyuan Yu, Zichen Kuang, Xinshu Qin, Hongxia Duan, Jie An

**Affiliations:** 1Department of Chemistry and Innovation Center of Pesticide Research, College of Science, China Agricultural University, Beijing 100193, China; lijunwang@cau.edu.cn (L.W.); lixiao23@mails.ucas.ac.cn (X.L.); yuzhiyuan27@163.com (Z.Y.); kuangzichen@126.com (Z.K.); 2Department of Nutrition and Health, China Agricultural University, Beijing 100083, China; lxchen915@163.com (L.C.); shxzq1@nottingham.edu.cn (Z.Q.); ke.ni@pepperdine.edu (K.N.); qinxs98@163.com (X.Q.)

**Keywords:** iodine, aerobic oxidation, disulfide

## Abstract

Iodine is a well-known oxidant that is widely used in organic syntheses. Thiol oxidation by stoichiometric iodine is one of the most commonly employed strategies for the synthesis of valuable disulfides. While recent advancements in catalytic aerobic oxidation conditions have eliminated the need for stoichiometric oxidants, concerns persist regarding the use of toxic or expensive catalysts. In this study, we discovered that iodine can be used as a cheap, low-toxicity catalyst in the aerobic oxidation of thiols. In the catalytic cycle, iodine can be regenerated via HI oxidation by O_2_ at 70 °C in EtOAc. This protocol harnesses sustainable oxygen as the terminal oxidant, enabling the conversion of primary and secondary thiols with remarkable efficiency. Notably, all 26 tested thiols, encompassing various sensitive functional groups, were successfully converted into their corresponding disulfides with yields ranging from >66% to 98% at a catalyst loading of 5 mol%.

## 1. Introduction

Disulfide bonds are a valuable functional group known for their remarkable biological significance. They are widely prevalent in various natural compounds, biological formulations, biomaterials, and pharmaceuticals in Figure 1 [1,2,3,4,5,6]. The oxidation of thiols offers a straightforward and viable approach for synthesizing disulfides. Traditionally, thiol oxidation has entailed the use of various stoichiometric oxidizing agents, including iodine, hydrogen peroxide, metal salts or metal oxides [7,8], halogens [9,10], high-valent sulfur oxidants [11,12,13], diethyl azodicarboxylate [14], etc. To avoid the use of stoichiometric oxidants, chemists are increasingly exploring catalytic oxidation methodologies that utilize oxygen [6]. Despite the fact that oxygen is attractive due to its natural abundance, cost-effectiveness, and eco-friendliness, its relatively low reactivity poses a challenge to its application in thiol oxidation [15,16,17,18]. As a result, various transition metal catalysts and organic catalysts have been developed for the aerobic oxidation of thiols. Transition metal catalysts used always contain organic ligands such as Co (II) phthalocyanines [19], Mn(III) Schiff-base complex [20], CoSalen [21], and Fe(Pc) [22]. Meanwhile, organic catalysts like tert-butyl nitrite [23], diisopropylamine [24], and TEMPO [25] suffer from difficulties in oxidizing secondary thiols. Recently, non-metallic inorganic catalysts like SiO_2_-Cl have also been developed, while the tedious preparation process has limited their applications [26].

Herein, we have established a novel I_2_-catalyzed aerobic oxidative thiol coupling strategy. I_2_, due to its readily available and low toxicity attributes, emerges as a highly suitable catalyst within environmentally benign processes [27,28,29,30]. The oxidation of thiols to disulfides using iodine has demonstrated a wide range of reactive groups tolerated [31]. Additionally, iodine has previously been employed as a catalyst in thiol oxidation combined with additives like DMSO, H_2_O_2_, and flavin [32,33,34]. This study showcases its effectiveness as a catalyst in catalyzing aerobic thiol oxidation at elevated temperatures using only 5 mol% of iodine, which is more cost-effective and environmentally friendly compared to previous studies. Notably, this green protocol exhibits good tolerance toward a diverse array of primary and secondary thiols bearing various functional groups.

## 2. Results and Discussion

This study commenced with reaction condition optimization of the iodine-catalyzed aerobic oxidation of thiols (Table 1). Dodecane-1-thiol **1a** was selected as the model substrate. The initial trial of the aerobic oxidation with 10 mol% I_2_ gave **2a** in >98% yield at 70 °C in EtOAc (Entry 1, Table 1). The high yield of **2a** was maintained when the amount of I_2_ was reduced to 5.0 mol% (Entry 2, Table 1). However, decreasing the catalyst loading of I_2_ to 1.0 mol% resulted in an incomplete conversion of **1a** (Entry 3, Table 1). The influence of reaction duration was investigated. Decreasing the reaction time from 4 h to 1 h resulted in a reduction in the reaction yield from >98% to 49%, which indicated extending the reaction time to 4 h can ensure the complete conversion of **1a** to **2a** (Entry 4, Table 1). Following the determination of optimal catalyst loading and reaction time, the impact of varying solvents on the reaction was explored. Substituting EtOAc with dichloromethane or *N*,*N*-dimethylformamide resulted in notably diminished yields (Entries 5 and 6, Table 1). This observation indicates the solvent’s pronounced influence on this catalytic reaction. Subsequently, the impact of temperature was examined. The findings revealed that the reaction conducted at room temperature (r.t.) yielded 53% of **2a** (Entry 7, Table 1), whereas the reaction conducted at 70 °C exhibited a significantly higher yield of >98%. Therefore, the optimal temperature for the experimental reaction was established at 70 °C. Finally, a control reaction with no catalyst was conducted. Significantly, only a trace amount of **2a** was formed in the control reaction, providing compelling evidence that I_2_ serves as an indispensable element for the synthesis of disulfide **2a** (Entry 8, Table 1).

Based on the comprehensive investigation of reaction conditions, we concluded that 5.0 mol% of I_2_ in EtOAc at 70 °C for a duration of 4 h was suitable for the substrate scope investigation (Figure 2). The substrate scope demonstrated that the I_2_-catalyzed aerobic oxidation protocol can convert all 23 tested primary and secondary thiols into disulfides in good to excellent yields. Notably, the aerobic oxidation process demonstrated a notable capacity to overcome the inherent challenges typically associated with secondary thiols, as exemplified by the good efficacy observed in the transformation of **1d** to **2d**. A variety of thiols with various functional groups were tested for the synthesis of symmetrical disulfides. Aryl thiols bearing both electron-withdrawing groups, including fluoride (**2e**, **2j**), chloride (**2f**, **2g**, and **2k**), and bromide (**2l**, **2m,** and **2n**), as well as electron-donating functional groups such as methoxyl (**2o**, **2p,** and **2q**), isopropyl (**2r**), methyl (**2s**), and amide (**2t**), afforded the corresponding disulfides in good to excellent yields. Extending the scope beyond the aforementioned substrates, the protocol effectively facilitated the conversion of ploy-aromatic (**2u**) and heteroaromatic (**2v** and **2w**) thiols into their respective disulfides. To underscore the practical utility of this approach, we have employed it in the oxidation of bioactive thiols, namely *N*-(*tert*-butoxycarbonyl)-*L*-cysteine methyl ester **1x** and *N*-acetyl-*L*-cystine **1y** (Figure 3), which have been used as treatments for acute paracetamol toxicity and peptide synthesis, respectively [35,36]. This resulted in the formation of the corresponding disulfides **2x** and **2y** in 66% and 98% yields, respectively. In addition, this method has also been applied to the oxidation of dithiol, dithiothreitol **1z** (Figure 4). The exclusive formation of the cyclized disulfide **2z** was achieved with an impressive yield of 98%, and there was no observed formation of the dimerized by-product. Notably, the formed trans-4,5-dihydroxy-1,2-dithiane **2z** is an inducer of ER stress proteins, which protects the kidney from chemical stress in vivo [37]. These results not only emphasize the method’s effectiveness but also highlight its potential for synthesizing intricate bioactive disulfides.

In order to explore the possible mechanism of this reaction, a series of control reactions was conducted in Figure 1. In the control reaction using 5 mol% HI to replace 5 mol% I_2_, **2a** was also formed in >98% yield (Figure 1(AI)). This result indicated that the catalyst iodine was regenerated from the oxidation of HI by oxygen. In the presence of TEMPO, a powerful free radical scavenger, the oxidation of thiols by stoichiometric iodine remained unaffected (Figure 1(AII,AIII)). Interestingly, TEMPO completely halted the iodine-catalyzed aerobic oxidation of thiols (Figure 1(AIV)). This observation strongly suggests that the oxidation of thiols might follow a distinct pathway within the catalytic cycle, contrasting with the oxidation process involving stoichiometric iodine.

## 3. Materials and Methods

### 3.1. General Information

Reagents and solvents were purchased from commercial suppliers and used directly without further purification, unless otherwise noted. All water was deionized before use. Unless otherwise noted, the glassware employed in the reactions was dried in an oven overnight before use. The oxygen purity used in the experiment is 99.999%.

NMR data were measured with a Bruker Avance NOE 500 and manipulated directly from the spectrometer or via a networked PC with appropriate software. Reference values for residual solvent were taken as δ = 7.27 (CDCl_3_) and δ = 2.50 (DMSO-*d*_6_) for ^1^H NMR; δ = 77.1 (CDCl_3_) and δ = 39.5 (DMSO-*d*_6_) for ^13^C{^1^H} NMR. Multiplicities for coupled signals were designated using the following abbreviations: s = singlet, d = doublet, t = triplet, q = quartet, quin = quintet, br = broad signal, and are given in Hz. Thin-layer chromatography was performed on SIL G/UV254 silica-glass plates, and the plates were visualized using ultra-violet light (254 nm) and KMnO_4_ solution. For flash column chromatography, silica gel (60, 35–70 μm) was used.

### 3.2. Calculation of the Yield by Internal Standard Using ^1^H NMR

The determination of yields by ^1^H NMR was according to the equation below:(1)$$\mathrm{Yield}=\left(\frac{{\mathrm{Area}}_{\mathrm{product}}}{{\mathrm{Area}}_{\mathrm{internal}\;\mathrm{standard}}}\right)\left(\frac{n_{\mathrm{internal}\;\mathrm{standard}}}{n_{\mathrm{theoretical}\;\mathrm{product}}}\right)\times 100\%$$

Area_product_ means the integration of the product peak; Area_internal standard_ means the integration of the internal standard peak; n_internal standard_ means the number of moles of the internal standard; n_theoretical product_ means the theoretical number of moles of the product.

### 3.3. Optimization Studies for the Oxidative Coupling of Thiols (Table 1)

To a round bottom flask were added I_2_ (0–7.61 mg, 0–10.0 mol%), dodecane-1-thiol (60.7 mg, 0.300 mmol, 1.00 equiv), and anhydrous EtOAc (8.00 mL). The flask was filled with an oxygen balloon (0.3 MPa), and the reaction mixture was stirred at 70 °C for a duration of 1–4 h. Subsequently, the reaction mixture was cooled to r.t. The reaction mixture was diluted with EtOAc (10.0 mL) and washed with HCl solution (15.0 mL, 0.100 M, aq). The aqueous layer was extracted with EtOAc (3 × 15.0 mL). Organic layers were combined, dried over MgSO_4_, filtered, and concentrated. The crude product was purified by flash chromatography (silica, 0–12.5% EtOAc/Hexane). The sample was then analyzed by ^1^H NMR (CDCl_3_, 500 MHz) to obtain the yield using the internal standard (1,1,2,2-tetrachloroethane) and comparison with corresponding samples.

### 3.4. General Procedure for the Oxidative Coupling of Thiols

To a round bottom flask were added I_2_ (3.81 mg, 5.00 mol%), thiol (0.300 mmol, 1.00 equiv), and EtOAc (8.00 mL). The flask was filled with an oxygen balloon (0.3 MPa), and the reaction mixture was stirred at 70 °C for a duration of 4 h. Subsequently, the reaction mixture was cooled to r.t. The reaction mixture was diluted with EtOAc (10.0 mL) and washed with HCl solution (15.0 mL, 0.100 M, aq). The aqueous layer was extracted with EtOAc (3 × 15.0 mL). Organic layers were combined, dried over MgSO_4_, filtered, and concentrated to give the crude product. The crude product was purified by flash chromatography (silica, 0–12.5% EtOAc/Hexane). Notably, ^1^H NMR and ^13^C{^1^H} NMR data of **2a**–**2z** were presented at Appendix A.

*1,2-Didodecyldisulfane* (**2a**) [38]. According to the general procedure, the oxidation of dodecane-1-thiol (60.7 mg, 0.300 mmol) catalyzed by I_2_ (3.81 mg, 5.00 mol%) under an oxygen balloon (0.3 MPa) after chromatography (100% Hexane), afforded 59.2 mg of **2a** in 98% yield as a colorless oil. ^1^H NMR (500 MHz, CDCl_3_) δ 2.68 (t, *J* = 7.4 Hz, 4H), 1.67 (m, 4H), 1.38 (m, 4H), 1.31–1.22 (m, 32H), 0.88 (t, *J* = 6.9 Hz, 6H); ^13^C{^1^H} NMR (126 MHz, CDCl_3_) δ 39.3, 32.0, 29.7 (×3), 29.6, 29.4, 29.3 (×2), 28.6, 22.8, 14.2.

*1,2-Diphenethyldisulfane* (**2b**) [39]. According to the general procedure, the oxidation of 2-phenylethane-1-thiol (41.5 mg, 0.300 mmol) catalyzed by I_2_ (3.81 mg, 5.00 mol%) under an oxygen balloon (0.3 MPa) after chromatography (0–12.5% EtOAc/Hexane) afforded 40.3 mg of **2b** in 98% yield as a yellow oil. ^1^H NMR (500 MHz, CDCl_3_) δ 7.29 (m, 4H), 7.23–7.16 (m, 6H), 3.01–2.95 (m, 4H), 2.95–2.90 (m, 4H); ^13^C{^1^H} NMR (126 MHz, CDCl_3_) δ 140.1, 128.7, 128.6, 126.5, 40.3, 35.8.

*1,2-Dibenzyldisulfane* (**2c**) [40]. According to the general procedure, the oxidation of phenylmethanethiol (37.3 mg, 0.300 mmol) catalyzed by I_2_ (3.81 mg, 5.00 mol%) under an oxygen balloon (0.3 MPa), after chromatography (0–10% EtOAc/Hexane), afforded 36.2 mg of **2c** in a 98% yield as a colorless oil. ^1^H NMR (500 MHz, CDCl_3_) δ 7.33–7.28 (m, 4H), 7.28–7.25 (m, 2H), 7.25–7.21 (m, 4H), 3.59 (s, 4H); ^13^C{^1^H} NMR (126 MHz, CDCl_3_) δ 137.4, 129.5, 128.6, 127.5, 43.4.

*1,2-Dicyclohexyldisulfane* (**2d**) [41]. According to the general procedure, the oxidation of cyclohexanethiol (34.9 mg, 0.300 mmol) catalyzed by I_2_ (3.81 mg, 5.00 mol%) under an oxygen balloon (0.3 MPa) after chromatography (100% Hexane) afforded 33.9 mg of **2d** in 98% yield as a colorless oil. ^1^H NMR (500 MHz, CDCl_3_) δ 2.68 (m, 2H), 2.12–1.94 (m, 4H), 1.86–1.71 (m, 4H), 1.67–1.52 (m, 2H), 1.39–1.16 (m, 10H); ^13^C{^1^H} NMR (126 MHz, CDCl_3_) δ 50.1, 32.9, 26.2, 25.8.

*1,2-Bis(4-fluorobenzyl)disulfane* (**2e**) [42]. According to the general procedure, the oxidation of (4-fluorophenyl)methanethiol (42.6 mg, 0.300 mmol) catalyzed by I_2_ (3.81 mg, 5.00 mol%) under an oxygen balloon (0.3 MPa) after chromatography (100% Hexane) afforded 41.5 mg of **2e** in 98% yield as a white solid. ^1^H NMR (500 MHz, CDCl_3_) δ 7.19 (m, 4H), 7.01 (m, 4H), 3.58 (s, 4H); ^13^C{^1^H} NMR (126 MHz, CDCl_3_) δ 162.3 (d, *J*_C-F_ = 246.4 Hz), 133.2 (d, *J*_C-F_ = 3.1 Hz), 131.0 (d, *J* _C-F_= 8.1 Hz), 115.5 (d, *J*_C-F_= 21.5 Hz), 42.5.

*1,2-Bis(4-chlorobenzyl)disulfane* (**2f**) [43]. According to the general procedure, the oxidation of (4-chlorophenyl)methanethiol (47.6 mg, 0.300 mmol) catalyzed by I_2_ (3.81 mg, 5.00 mol%) under an oxygen balloon (0.3 MPa) after chromatography (100% Hexane) afforded 46.3 mg of **2f** in 98% yield as a white solid. ^1^H NMR (500 MHz, CDCl_3_) δ 7.29 (m, 4H), 7.15 (m, 4H), 3.57 (s, 4H); ^13^C{^1^H} NMR (126 MHz, CDCl_3_) δ 135.9, 133.5, 130.7, 128.7, 42.6.

*1,2-Bis(2-chlorobenzyl)disulfane* (**2g**) [38]. According to the general procedure, the oxidation of (2-chlorophenyl)methanethiol (47.6 mg, 0.300 mmol) catalyzed by I_2_ (3.81 mg, 5.00 mol%) under an oxygen balloon (0.3 MPa) after chromatography (0–12.5% EtOAc/Hexane) afforded 46.3 mg of **2g** in 98% yield as a white solid. ^1^H NMR (500 MHz, CDCl_3_) δ 7.37 (m, 2H), 7.26 (m, 2H), 7.24–7.20 (m, 4H), 3.78 (s, 4H); ^13^C{^1^H} NMR (126 MHz, CDCl_3_) δ 135.1, 134.2, 131.7, 129.8, 129.0, 126.8, 41.2.

*1,2-Bis(4-methoxybenzyl)disulfane* (**2h**) [38]. According to the general procedure, the oxidation of (4-methoxyphenyl)methanethiol (46.3 mg, 0.300 mmol) catalyzed by I_2_ (3.81 mg, 5.00 mol%) under an oxygen balloon (0.3 MPa) after chromatography (0–12.5% EtOAc/Hexane) afforded 45.0 mg of **2h** in 98% yield as a white solid. ^1^H NMR (500 MHz, CDCl_3_) δ 7.16 (m, 4H), 6.85 (m, 4H), 3.79 (s, 6H), 3.59 (s, 4H); ^13^C{^1^H} NMR (126 MHz, CDCl_3_) δ 159.1, 130.6, 129.5, 114.0, 55.3, 42.8.

*1,2-Bis(4-(tert-butyl)benzyl)disulfane* (**2i**) [42]. According to the general procedure, the oxidation of (4-(*tert*-butyl)phenyl)methanethiol (54.1 mg, 0.300 mmol) catalyzed by I_2_ (3.81 mg, 5.00 mol%) under an oxygen balloon (0.3 MPa) after chromatography (0–12.5% EtOAc/Hexane) afforded 52.7 mg of **2i** in 98% yield as a colorless oil. ^1^H NMR (500 MHz, CDCl_3_) δ 7.33 (m, 4H), 7.17 (m, 4H), 3.59 (s, 4H), 1.31 (s, 18H); ^13^C{^1^H} NMR (126 MHz, CDCl_3_) δ 150.5, 134.3, 129.2, 125.5, 43.1, 34.6, 31.4.

*1,2-Bis(4-fluorophenyl)disulfane* (**2j**) [40]. According to the general procedure, the oxidation of 4-fluorobenzenethiol (38.4 mg, 0.300 mmol) catalyzed by I_2_ (3.81 mg, 5.00 mol%) under an oxygen balloon (0.3 MPa) after chromatography (100% Hexane) afforded 37.4 mg of **2j** in 98% yield as a white solid. ^1^H NMR (500 MHz, CDCl_3_) δ 7.43 (m, 4H), 7.00 (m, 4H); ^13^C{^1^H} NMR (126 MHz, CDCl_3_) δ 162.7 (d, *J*_C-F_ = 247.8 Hz), 132.3 (d, *J*_C-F_ = 3.2 Hz), 131.4 (d, *J*_C-F_ = 8.1 Hz), 116.4 (d, *J*_C-F_ = 22.5 Hz).

*1,2-Bis(4-chlorophenyl)disulfane* (**2k**) [40]. According to the general procedure, the oxidation of 4-chlorobenzenethiol (43.4 mg, 0.300 mmol) catalyzed by I_2_ (3.81 mg, 5.00 mol%) under an oxygen balloon (0.3 MPa) after chromatography (100% Hexane) afforded 42.2 mg of **2k** in 98% yield as a white solid. ^1^H NMR (500 MHz, CDCl_3_) δ 7.39 (m, 4H), 7.27 (m, 4H); ^13^C{^1^H} NMR (126 MHz, CDCl_3_) δ 135.2, 133.7, 129.4 (×2).

*1,2-Bis(4-bromophenyl)disulfane* (**2l**) [40]. According to the general procedure, the oxidation of 4-bromobenzenethiol (56.7 mg, 0.300 mmol) catalyzed by I_2_ (3.81 mg, 5.00 mol%) under an oxygen balloon (0.3 MPa) after chromatography (100% Hexane) afforded 47.4 mg of **2l** in 84% yield as a white solid. ^1^H NMR (500 MHz, CDCl_3_) δ 7.41 (m, 4H), 7.32 (m, 4H); ^13^C{^1^H} NMR (126 MHz, CDCl_3_) δ 135.8, 132.3, 129.4, 121.6.

*1,2-Bis(3-bromophenyl)disulfane* (**2m**) [39]. According to the general procedure, the oxidation of 3-bromobenzenethiol (56.7 mg, 0.300 mmol) catalyzed by I_2_ (3.81 mg, 5.00 mol%) under an oxygen balloon (0.3 MPa) after chromatography (100% Hexane) afforded 55.3 mg of **2m** in a 98% yield as a colorless oil. ^1^H NMR (500 MHz, CDCl_3_) δ 7.62 (m, 2H), 7.41–7.33 (m, 4H), 7.17 (m, 2H); ^13^C{^1^H} NMR (126 MHz, CDCl_3_) δ 138.7, 130.6, 130.5, 130.0, 126.0, 123.2.

*1,2-Bis(2-bromophenyl)disulfane* (**2n**) [41]. According to the general procedure, the oxidation of 2-bromobenzenethiol (56.7 mg, 0.300 mmol) catalyzed by I_2_ (3.81 mg, 5.00 mol%) under an oxygen balloon (0.3 MPa) after chromatography (100% Hexane) afforded 53.6 mg of **2n** in a 95% yield as a white solid. ^1^H NMR (500 MHz, CDCl_3_) δ 7.55–7.50 (m, 4H), 7.26 (m, 2H), 7.07 (m, 2H); ^13^C{^1^H} NMR (126 MHz, CDCl_3_) δ 136.2, 133.0, 128.3, 128.0, 127.0, 121.1.

*1,2-Bis(4-methoxyphenyl)disulfane* (**2o**) [40]. According to the general procedure, the oxidation of 4-methoxybenzenethiol (42.1 mg, 0.300 mmol) catalyzed by I_2_ (3.81 mg, 5.00 mol%) under an oxygen balloon (0.3 MPa) after chromatography (0–12.5% EtOAc/Hexane) afforded 40.9 mg of **2o** in 98% yield as a yellow oil. ^1^H NMR (500 MHz, CDCl_3_) δ 7.39 (m, 4H), 6.83 (m, 4H), 3.79 (s, 6H); ^13^C{^1^H} NMR (126 MHz, CDCl_3_) δ 160.0, 132.7, 128.5, 114.7, 55.4.

*1,2-Bis(2-methoxyphenyl)disulfane* (**2p**) [40]. According to the general procedure, the oxidation of 2-methoxybenzenethiol (42.1 mg, 0.300 mmol) catalyzed by I_2_ (3.81 mg, 5.00 mol%) under an oxygen balloon (0.3 MPa) after chromatography (0–12.5% EtOAc/Hexane), afforded 40.9 mg of **2p** in 98% yield as a white solid. ^1^H NMR (500 MHz, CDCl_3_) δ 7.53 (m, 2H), 7.17 (m, 2H), 6.90 (m, 2H), 6.84 (m, 2H), 3.88 (s, 6H); ^13^C{^1^H} NMR (126 MHz, CDCl_3_) δ 156.7, 127.8, 127.7, 124.6, 121.4, 110.6, 55.9.

*1,2-Bis(3,4-dimethoxyphenyl)disulfane* (**2q**) [40]. According to the general procedure, the oxidation of 3,4-dimethoxybenzenethiol (51.1 mg, 0.300 mmol) catalyzed by I_2_ (3.81 mg, 5.00 mol%) under an oxygen balloon (0.3 MPa) after chromatography (0–12.5% EtOAc/Hexane) afforded 49.8 mg of **2q** in 98% yield as a white solid. ^1^H NMR (500 MHz, CDCl_3_) δ 7.06 (d, *J* = 2.1 Hz, 1H), 7.04 (d, *J* = 2.1 Hz, 1H), 7.01 (m, 2H), 6.79 (s, 1H), 6.78 (s, 1H), 3.87 (s, 6H), 3.83 (s, 6H); ^13^C{^1^H} NMR (126 MHz, CDCl_3_) δ 149.6, 149.2, 128.7, 123.9, 114.1, 111.3, 56.0, 55.9.

*1,2-Bis(4-isopropylphenyl)disulfane* (**2r**) [44]. According to the general procedure, the oxidation of 4-isopropylbenzenethiol (45.6 mg, 0.300 mmol) catalyzed by I_2_ (3.81 mg, 5.00 mol%) under an oxygen balloon (0.3 MPa) after chromatography (100% Hexane) afforded 39.0 mg of **2r** in 86% yield as a colorless oil. ^1^H NMR (500 MHz, CDCl_3_) δ 7.42 (m, 4H), 7.16 (m, 4H), 2.87 (m, 2H), 1.22 (d, *J* = 6.9 Hz, 12H); ^13^C{^1^H} NMR (126 MHz, CDCl_3_) δ 148.4, 134.4, 128.3, 127.3, 33.8, 24.0.

*1,2-Di-p-tolyldisulfane* (**2s**) [40]. According to the general procedure, the oxidation of 4-methylbenzenethiol (37.3 mg, 0.300 mmol) catalyzed by I_2_ (3.81 mg, 5.00 mol%) under an oxygen balloon (0.3 MPa) after chromatography (100% Hexane) afforded 32.2 mg of **2s** in 87% yield as a white solid. ^1^H NMR (500 MHz, CDCl_3_) δ 7.38 (m, 4H), 7.10 (m, 4H), 2.32 (s, 6H); ^13^C{^1^H} NMR (126 MHz, CDCl_3_) δ 137.5, 134.0, 129.9, 128.6, 21.1.

*N,N′-(Disulfanediylbis(4,1-phenylene))diacetamide* (**2t**) [41]. According to the general procedure, the oxidation of *N*-(4-mercaptophenyl)acetamide (50.2 mg, 0.300 mmol) catalyzed by I_2_ (3.81 mg, 5.00 mol%) under an oxygen balloon (0.3 MPa) after chromatography (0–10% MeOH/EtOAc) afforded 40.4 mg of **2t** in 81% yield as a white solid. ^1^H NMR (500 MHz, DMSO-*d*_6_) δ 10.07 (s, 2H), 7.59 (m, 4H), 7.42 (m, 4H), 2.04 (s, 6H); ^13^C{^1^H} NMR (126 MHz, DMSO-*d_6_*) δ 168.5, 139.5, 130.1, 129.4, 119.7, 24.0.

*1,2-Di(naphthalen-2-yl)disulfane* (**2u**) [40]. According to the general procedure, the oxidation of naphthalene-2-thiol (48.1 mg, 0.300 mmol) catalyzed by I_2_ (3.81 mg, 5.00 mol%) under an oxygen balloon (0.3 MPa) after chromatography (100% Hexane) afforded 46.8 mg of **2u** in 98% yield as a white solid. ^1^H NMR (500 MHz, CDCl_3_) δ 7.97 (m, 2H), 7.79–7.74 (m, 4H), 7.71 (m, 2H), 7.61 (m, 2H), 7.48–7.40 (m, 4H); ^13^C{^1^H} NMR (126 MHz, CDCl_3_) δ 134.3, 133.6, 132.6, 129.1, 127.8, 127.5, 126.8, 126.6, 126.3, 125.7.

*1,2-Di(thiophen-2-yl)disulfane* (**2v**) [41]. According to the general procedure, the oxidation of thiophene-2-thiol (34.9 mg, 0.300 mmol) catalyzed by I_2_ (3.81 mg, 5.00 mol%) under an oxygen balloon (0.3 MPa) after chromatography (100% Hexane) afforded 28.3 mg of **2v** in 82% yield as a colorless oil. ^1^H NMR (500 MHz, CDCl_3_) δ 7.48 (d, *J* = 5.2 Hz, 2H), 7.14 (d, *J* = 3.7 Hz, 2H), 7.00 (dd, *J* = 5.2, 3.7 Hz, 2H); ^13^C{^1^H} NMR (126 MHz, CDCl_3_) δ 135.8, 135.7, 132.3, 127.8.

*1,2-Bis(furan-2-ylmethyl)disulfane* (**2w**) [39]. According to the general procedure, the oxidation of furan-2-ylmethanethiol (34.2 mg, 0.300 mmol) catalyzed by I_2_ (3.81 mg, 5.00 mol%) under an oxygen balloon (0.3 MPa) after chromatography (0–12.5% EtOAc/Hexane) afforded 33.3 mg of **2w** in 98% yield as a colorless oil. ^1^H NMR (500 MHz, CDCl_3_) δ 7.38 (m, 2H), 6.33 (dd, *J* = 3.2, 2.0 Hz, 2H), 6.22 (d, *J* = 3.2 Hz, 2H), 3.68 (s, 4H); ^13^C{^1^H} NMR (126 MHz, CDCl_3_) δ 150.3, 142.5, 110.8, 109.0, 35.7.

*Dimethyl 3,3′-disulfanediyl(2R,2′R)-bis(2-((tert-butoxycarbonyl)amino)propanoate*) (**2x**) [45]. According to the general procedure, the oxidation of *N*-(*tert*-butoxycarbonyl)-*L*-cysteine methyl ester (70.6 mg, 0.300 mmol) catalyzed by I_2_ (3.81 mg, 5.00 mol%) under an oxygen balloon (0.3 MPa) after chromatography (0–30% EtOAc/Hexane) afforded 46.4 mg of **2y** in 66% yield as a white solid. ^1^H NMR (500 MHz, DMSO-*d*_6_) δ 7.36 (d, *J* = 8.2 Hz, 2H), 4.26 (m, 2H), 3.64 (s, 6H), 3.07 (m, 2H), 2.90 (m, 2H), 1.37 (s, 18H); ^13^C{^1^H} NMR (126 MHz, CDCl_3_) δ 171.4, 155.3, 78.6, 52.7, 52.1, 39.1, 28.1.

*(2R,2′R)-3,3′-disulfanediylbis(2-acetamidopropanoic acid)* (**2y**) [20]. According to the general procedure, the oxidation of *N*-acetyl-*L*-cysteine (49.0 mg, 0.300 mmol) is catalyzed by I_2_ (3.81 mg, 5.00 mol%) under an oxygen balloon (0.3 MPa). The reaction mixture was concentrated, then washed with EtOAc (20 mL), affording 47.7 mg of **2z** in 98% yield as a white solid. ^1^H NMR (500 MHz, D_2_O) δ 4.68 (dd, *J* = 8.6, 4.3 Hz, 2H), 3.38 (dd, *J* = 14.1, 4.3 Hz, 2H), 3.02 (dd, *J* = 14.1, 8.6 Hz, 2H), 2.04 (s, 6H); ^13^C{^1^H} NMR (126 MHz, D_2_O) δ 177.7, 174.1, 53.2, 39.4, 21.8.

*(4R,5R)-1,2-Dithiane-4,5-diol* (**2z**) [46]. According to the general procedure, the oxidation of (2*R*,3*R*)-1,4-dimercaptobutane-2,3-diol (46.3 mg, 0.300 mmol) catalyzed by I_2_ (3.81 mg, 5.00 mol%) under an oxygen balloon (0.3 MPa) after chromatography (0–100% EtOAc/Hexane) afforded 44.8 mg of **2z** in 98% yield as a white solid. ^1^H NMR (500 MHz, CD_3_OD) δ 3.48–3.31 (m, 2H), 3.02–2.89 (m, 2H), 2.84–2.73 (m, 2H); ^13^C{^1^H} NMR (126 MHz, CD_3_OD) δ 74.09, 40.4.

### 3.5. Procedure for Control Experiments (Figure 1)

Figure 1(AI): To a round bottom flask were added 55 wt% HI (3.49 mg, 5.00 mol%), dodecane-1-thiol (60.7 mg, 0.300 mmol, 1.00 equiv), and anhydrous EtOAc (8.00 mL). The flask was filled with an oxygen balloon (0.3 MPa), and the reaction mixture was stirred at 70 °C for a duration of 4 h. Subsequently, the reaction mixture was cooled to r.t. The reaction mixture was diluted with EtOAc (10.0 mL) and washed with HCl solution (15.0 mL, 0.100 M, aq). The aqueous layer was extracted with EtOAc (3 × 15.0 mL). Organic layers were combined, dried over MgSO_4_, filtered, and concentrated. The crude product was then analyzed by ^1^H NMR (CDCl_3_, 500 MHz) to obtain the yield using the internal standard (1,1,2,2-tetrachloroethane) and comparison with corresponding samples.

Figure 1(AII): To a round bottom flask were added I_2_ (38.1mg, 50.0 mol%), dodecane-1-thiol (60.7 mg, 0.300 mmol, 1.00 equiv), and anhydrous EtOAc (8.00 mL). The flask was filled with an oxygen balloon (0.3 MPa), and the reaction mixture was stirred at 70 °C for a duration of 4 h. Subsequently, the reaction mixture was cooled to r.t. The reaction mixture was diluted with EtOAc (10.0 mL) and washed with HCl solution (15.0 mL, 0.100 M, aq). The aqueous layer was extracted with EtOAc (3 × 15.0 mL). Organic layers were combined, dried over MgSO_4_, filtered, and concentrated. The crude product was then analyzed by ^1^H NMR (CDCl_3_, 500 MHz) to obtain the yield using the internal standard (1,1,2,2-tetrachloroethane) and comparison with corresponding samples.

Figure 1(AIII): To a round bottom flask were added I_2_ (38.1 mg, 50.0 mol%), dodecane-1-thiol (60.7 mg, 0.300 mmol, 1.00 equiv), TEMPO (1.00 equiv), and anhydrous EtOAc (8.00 mL). The flask was filled with an oxygen balloon (0.3 MPa), and the reaction mixture was stirred at 70 °C for a duration of 4 h. Subsequently, the reaction mixture was cooled to r.t. The reaction mixture was diluted with EtOAc (10.0 mL) and washed with HCl solution (15.0 mL, 0.100 M, aq). The aqueous layer was extracted with EtOAc (3 × 15.0 mL). Organic layers were combined, dried over MgSO_4_, filtered, and concentrated. The crude product was then analyzed by ^1^H NMR (CDCl_3_, 500 MHz) to obtain the yield using the internal standard (1,1,2,2-tetrachloroethane) and comparison with the corresponding sample.

Figure 1(AIV): To a round bottom flask were added I_2_ (3.81 mg, 5.00 mol%), dodecane-1-thiol (60.7 mg, 0.300 mmol, 1.00 equiv), TEMPO (1.00 equiv), and anhydrous EtOAc (8.00 mL). The flask was filled with an oxygen balloon (0.3 MPa), and the reaction mixture was stirred at 70 °C for a duration of 4 h. Subsequently, the reaction mixture was cooled to r.t. The reaction mixture was diluted with EtOAc (10.0 mL) and washed with HCl solution (15.0 mL, 0.100 M, aq). The aqueous layer was extracted with EtOAc (3 × 15.0 mL). Organic layers were combined, dried over MgSO_4_, filtered, and concentrated. The crude product was then analyzed by ^1^H NMR (CDCl_3_, 500 MHz) to obtain the yield using the internal standard (1,1,2,2-tetrachloroethane) and comparison with corresponding samples.

## 4. Conclusions

In summary, we have established a cost-effective and environmentally friendly I_2_-catalyzed aerobic oxidative coupling of thiols for the synthesis of valuable disulfide. In contrast to reported catalytic aerobic oxidation methods, this protocol circumvented the need for transition-metal catalysts and reagents that are not commercially available. This novel method tolerates both primary and secondary alkyl thiols, as well as aryl thiols. All 26 tested substrates with various functional groups resulted in good yields, which highlighted the exceptional functional group compatibility of this approach. This sustainable methodology holds promise for widespread applicability across both academic and industrial realms.

## Data Availability

Not applicable.

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
