# Peer review of "Application of Iodine as a Catalyst in Aerobic Oxidations: A Sustainable Approach for Thiol Oxidations"

_molecules, 2023, doi:10.3390/molecules28196789_

Round 1

Reviewer 1 Report

In the submitted manuscript, the authors have reported the aerobic oxidative dimerization of thiols to disulfides using iodine as a catalyst. However, I would like to express some concerns regarding the introduction section, specifically concerning the presentation of related literature. The statement, “While iodine has previously been employed as a stoichiometric oxidant in thiol oxidation,” is not entirely accurate. Kirihara et al. have indeed reported the iodine-catalyzed oxidative dimerization of thiols to disulfides using hydrogen peroxide as an oxidant (Synthesis 2007, 3286. 10.1055/s-2007-990800). Additionally, Iida et al. have documented the aerobic oxidation of thiols using flavin/iodine co-catalysis (Synlett 2021, 32, 1227. 10.1055/a-1520-9916). I recommend revising the introduction to provide a more precise account of the prior research in this field and clearly elucidate how the authors’ study distinguishes itself or extends beyond these earlier works. Emphasizing the distinctive contributions of the present results compared to existing literature will undoubtedly enhance the overall impact of the paper.

Furthermore, this reviewer suggests including "I2 (catalyst)" on the reaction arrow in Table 1 for clarity. Additionally, in the experimental section, please specify the method used for oxygen introduction, whether it was via O2 flow, an O2 balloon, or a sealed system under O2 pressure.

I strongly recommend making these revisions before considering this manuscript for publication.

Author Response

We thank reviewer 1 for the comments. In revised manuscript, author have now summarized the prior research in iodine-catalyzed reactions in the introduction and emphasized the distinctive contributions of our work compared with previous research. In addition, "I2 (catalyst)" has been added to the reaction arrow in Table 1 and the use of oxygen has been supplemented in the experimental section.

Reviewer 2 Report

      Iodine is a widely used oxidant in organic synthesis, particularly in thiol oxidation for valuable disulfides. This study found that iodine can be used as a cheap, low-toxicity catalyst in aerobic thiol oxidations. The process uses sustainable oxygen as the terminal oxidant, enabling efficient conversion of primary and secondary thiols with yields of 66% to 98%.

     Manuscripts have some grammatical and language errors which need to be eliminated prior to being accepted for publication. Here are my comments. 

1.     What is the specific reason for using EtOAc as the preferred solvent?

2.     The introduction section lacks a literature review highlighting the significance of aerobic oxidation of thiols and iodine for this particular research. Suggested to emphasizes the hazards connected with iodine in comparison to non-iodine substitutes.

3.     Room temperature is a common term and quite understandable. No need to mention the specific temperature range.

4.     Line 153. What means by “Extra dry”. Or it means anhydrous?

5.     Line 154. When filling the flask with oxygen, what pressure is used, and what is the gas's purity level?

6.     For the conversion to disulfides, this investigation documents 26 thiols. Does it involve any sort of analysis-based optimization?

7.     While decreasing the reaction time from 4h to 1h, there is a sharp decline in yield from 98% to 49%. Why is this decline taking place?

Author Response

  1. What is the specific reason for using EtOAc as the preferred solvent?

The authors’ reply: We thank reviewer 2 for the comments. Compared to EtOAc, DMF is difficult to remove and DCM is not environmentally friendly. Under the same reaction conditions, it cannot effectively convert thiol to disulfide in these two solvents. Additionally, EtOAc demonstrated excellent solubility for our tested substrate, which facilitates the progress of the reaction.

  1. The introduction section lacks a literature review highlighting the significance of aerobic oxidation of thiols and iodine for this particular research. Suggested to emphasizes the hazards connected with iodine in comparison to non-iodine substitutes.

The authors’ reply: We thank reviewer 2 for the comments. we have summarized the importance of aerobic oxidation of thiols and iodine for this particular research and the disadvantage of the prior research in iodine-catalyzed reaction.

  1. Room temperature is a common term and quite understandable. No need to mention the specific temperature range.

The authors’ reply: We thank reviewer 2 for the comments. In revised manuscript, we have deleted in line 135.

  1. Line 153. What means by “Extra dry”. Or it means anhydrous?

The authors’ reply: We thank reviewer 2 for the comments. The ethyl acetate was used in our experiment to remove water as much as possible, its water content below 50 ppm, and we have revised in revised manuscript.

  1. Line 154. When filling the flask with oxygen, what pressure is used, and what is the gas's purity level?

The authors’ reply: We thank reviewer 2 for the comments. Oxygen balloon (0.3 MPa) is applied in reaction and the oxygen purity used is 99.999%, which have been supplemented in revised manuscript.

  1. For the conversion to disulfides, this investigation documents 26 thiols. Does it involve any sort of analysis-based optimization?

The authors’ reply: We thank reviewer 2 for the comments. The analysis-based optimization was not applied to these 26 substrates including primary and secondary thiols. In future, we will employ analysis-based optimization to explore substrate scope in organic synthesis.

  1. While decreasing the reaction time from 4h to 1h, there is a sharp decline in yield from 98% to 49%. Why is this decline taking place?

The authors’ reply: We thank reviewer 2 for the comments. In this work, turnover number of iodine-catalyzed reaction is 20 according to the amount of catalyst, sufficient time is required for the complete catalytic cycle. Therefore, there is a sharp decline in yield from 98% to 49% when the reaction time was reduced from 4h to 1h.

Reviewer 3 Report

The authors submitted a manuscript dedicated to the use of iodine as catalyst in the aerobic oxidation of thiols with biological significance claiming that iodine is not a toxic or expensive catalyst.

The main drawback of the submitted material is related to the description of the experiments, which makes impossible the reproduction in other laboratories.

In section 3.4 General Procedure for the Oxidative Coupling of Thiols

First of all the authors do not state the volume of the round bottom flask used in the experiments, they do not state how they calculated  I2 molar percentage (if iodine is added as I2 solid crystals 3.79 mg represent ca 9.05 mol% calculated with respect to the total number of mmols of I2 and the respective thiol, if the percentage is calculated with respect of the total amount of I2, thiol and EtOAc the molar percentage of I2 is 7.25 %, both percentages are far from 5% claimed by the authors !), and finally they do not state the pressure of oxygen.

In section 3.5. Procedure of control experiments (Scheme 1)

line 319 To a round bottom flask was added HI (5.00 mol %), - what amount (mg, mL?) How was the percentage calculated?

same questions for lines 328 and 337 "To a round bottom flask was added I2 (50.0 mol %)"

and for line 346 "To a round bottom flask was added I2 (5.00 mol %),"

The discussion of the reaction mechanism presented at lines 118-127 should be extended and clarified. The authors should explain why the presence of TEMPO at room temperature in the absence of O2 increased the conversion while at 70oC in the presence of O2 it inhibited the reaction.

At line 59-60

However, decreasing the catalyst loading of I2 to 1.0 mol % resulted in incomplete conversion of 21a (Entry 3, Table 1) (I think that it should be 1a instead of 2a which is the product!

In the abstract

"Iodine is a well-known oxidant that is widely used in organic synthesises. Among them, tThiol oxidation by stoichiometric iodine is one of the mostly employed strategy for the synthesis of valuable disulfides.

line 21

encompassing virous various sensitive

line 31 metal salts or metal oxides

lines 30-41

While Meanwhile, organic catalysts like tert-butyl nitrite [23], diisopropylamine [24] and TEMPO [25] suffer from difficulties in oxidizing secondary thiols.

line 87: The substrate scope investigation demonstrated that ( rephrase)

at lines 89, 107 use Notably instead of "Of note"

line 128 Scheme 1 Experimental evidence and proposed mechanism

try to give a better title may be "Experimental strategy for determining the reaction mechanism"

Author Response

  1. In section 3.4 General Procedure for the Oxidative Coupling of Thiols

First of all the authors do not state the volume of the round bottom flask used in the experiments, they do not state how they calculated I2 molar percentage (if iodine is added as I2 solid crystals 3.79 mg represent ca 9.05 mol% calculated with respect to the total number of mmols of I2 and the respective thiol, if the percentage is calculated with respect of the total amount of I2, thiol and EtOAc the molar percentage of I2 is 7.25 %, both percentages are far from 5% claimed by the authors !), and finally they do not state the pressure of oxygen.

The authors’ reply: We thank reviewer 3 for the comments. In catalytic reaction, the molar percentage of catalyst is obtained in comparison to the substrate (Asian J. Org. Chem. 2017, 6, 265–268; Green Chem. 2021, 23, 546–551). The ratio of I2 as catalyst to thiol as reactant is 5 mol% and we have revised the amount of I2. Besides, oxygen balloon (0.3 MPa) was applied in this work, which has been supplemented in revised manuscript.

  1. In section 3.5. Procedure of control experiments (Scheme 1)

line 319: To a round bottom flask was added HI (5.00 mol %), - what amount (mg, mL?) How was the percentage calculated? same questions for lines 328 and 337 "To a round bottom flask was added I2 (50.0 mol %,)" and for line 346 "To a round bottom flask was added I2 (5.00 mol %),"

The authors’ reply: We thank reviewer 3 for the comments. The percentage was calculated by the ratio of HI or I2 to thiol and the detailed amount of HI and I2 has been supplemented in section 3.5.

  1. The discussion of the reaction mechanism presented at lines 118-127 should be extended and clarified. The authors should explain why the presence of TEMPO at room temperature in the absence of O2 increased the conversion while at 70oC in the presence of O2 it inhibited the reaction.

The authors’ reply: We thank reviewer 3 for the comments. At lines 118-127, we have mentioned that this reaction using catalytic iodine differs from stoichiometric iodine reaction. Actually, there is no obvious effect in yield in the presence and absence of TEMPO at room temperature, which is within the permissible range of error in yield using internal standard calculation.

  1. At line 59-60: However, decreasing the catalyst loading of I2 to 1.0 mol % resulted in incomplete conversion of 21a (Entry 3, Table 1) (I think that it should be 1a instead of 2a which is the product!

The authors’ reply: We thank reviewer 3 for the comments. We have checked in revised manuscript.

In the abstract, "Iodine is a well-known oxidant that is widely used in organic synthesises. Among them, tThiol oxidation by stoichiometric iodine is one of the mostly employed strategy for the synthesis of valuable disulfides.

The authors’ reply: We thank reviewer 3 for the comments. We have checked in revised manuscript.

  1. line 21: encompassing virous various sensitive

The authors’ reply: We thank reviewer 3 for the comments. We have checked in revised manuscript.

  1. line 31: metal salts or metal oxides

The authors’ reply: We thank reviewer 3 for the comments. We have checked in revised manuscript.

  1. lines 30-41: While Meanwhile, organic catalysts like tert-butyl nitrite [23], diisopropylamine [24] and TEMPO [25] suffer from difficulties in oxidizing secondary thiols.

The authors’ reply: We thank reviewer 3 for the comments. We have checked in revised manuscript.

  1. line 87: The substrate scope investigation demonstrated that (rephrase)

The authors’ reply: We thank reviewer 3 for the comments. We have changed the the substrate scope investigation the substrate scope in revised manuscript.

  1. At lines 89, 107: use Notably instead of "Of note"

The authors’ reply: We thank reviewer 3 for the comments. We have changed "Of note" to Notably in revised manuscript.

  1. line 128: “Scheme 1 Experimental evidence and proposed mechanism” try to give a better title may be "Experimental strategy for determining the reaction mechanism"

The authors’ reply: We thank reviewer 3 for the comments. We have changed in revised manuscript.

Round 2

Reviewer 3 Report

The authors revised the manuscript and answered to all my queries.

Author Response

We thank reviewer 3 for the comments, we have added a reference in revised manuscript.